# Structural determinants of inequalities in untreated dental caries in the Global Burden of Disease Study

Orlando Luiz do Amaral Junior[1], Maria Laura Braccini Fagundes[1], Fernando Neves Hugo[2]*, Nicholas J. Kassebaum[3], Jessye Melgarejo do Amaral Giordani[1]

1 Postgraduate Program in Dental Sciences, Federal University of Santa Maria, Santa Maria, Brazil, 2 College of Dentistry, Department of Epidemiology and Health Promotion, New York University, New York, New York, United States of America, 3 Institute for Health Metrics and Evaluation, University of Washington, Seattle, United States of America

* fnh9064@nyu.edu

## Abstract

### Introduction

Untreated dental caries remains a significant global public health challenge, disproportionately affecting vulnerable populations and highlighting inequalities in health systems worldwide. Examining its distribution across diverse contexts is essential for guiding targeted interventions, informing public policies, and allocating resources more effectively. This study aims to provide a comprehensive global assessment of the distribution of untreated dental caries in 204 countries and territories, spanning the years 2000, 2010, and 2019, by dimensions of structural determinants of health.

### Methods

This study employs an ecological design. This analysis describes the incidence, prevalence and years lived with disability (YLDs) due to untreated dental caries in permanent and deciduous teeth by structural determinants of health (dimensions of governance, macroeconomic policy, social policy, public policies and culture/societal values), for 204 countries and territories. Estimates of untreated caries in deciduous teeth by structural determinants were reported for children under 5 years of age, whereas estimates in permanent teeth were described for adolescents and adults aged between 15 and 49 years. Estimates were retrieved from the Global Health Data Exchange.

### Results

There was a gradient of inequality between countries in terms of prevalence, YLDs, and the incidence of overt untreated caries. Countries that had better structural indicators had lower mean prevalence of untreated caries in primary teeth. In the

**Data availability statement:** The data underlying the results presented in the study are available from the Institute for Health Metrics and Evaluation (IHME) and the World Bank. This study exclusively utilized secondary, anonymized data from the Global Burden of Disease (GBD) study and the World Bank database, both of which are publicly accessible and do not require ethical approval for use. In accordance with the policies established by the IHME, all data provided through the GBD initiative are aggregated and anonymized to ensure individual privacy and adherence to ethical research standards. Similarly, the World Bank database offers open access to global development indicators and anonymized datasets for research and public use. More information on data access policies and ethical guidelines can be found at the IHME website (https://www.healthdata.org) and the World Bank Open Data portal (https://data.worldbank.org).

**Funding:** F.N.H. received research funding from New York University (NYU), which supported his contribution to this work. O.L.A.J. received financial support from CAPES (Coordenação de Aperfeiçoamento de Pessoal de Nível Superior), a Brazilian research funding agency. The other authors did not receive specific funding.

**Competing interests:** The authors have declared that no competing interests exist.

permanent dentition, between-countries inequalities were not identified. Additionally, the mean prevalence was stable from 2010 to 2019.

## Conclusions

This study revealed possible cross-country inequalities in the burden of untreated caries in deciduous teeth that persisted over one decade. More developed countries experienced less disease burden. These findings show the need for targeted interventions to address the uneven burden of untreated dental caries worldwide.

## Clinical significance

This study underscores the need for targeted global health policies to reduce untreated dental caries, especially in countries with lower structural health determinants, highlighting cross-country inequalities in disease burden and the importance of addressing these disparities in both primary and permanent dentition.

## Introduction

Despite efforts to tackle dental caries over the past 50 years, a significant proportion of the world's population continues to suffer from the disease [1,2]. Caries have been identified by the World Health Organization as an urgent public health challenge with social, economic, and environmental impacts [3]. Unfortunately, poor, vulnerable, and socially disadvantaged individuals and groups are disproportionately affected by it [3]. Therefore, it's crucial for governments and societies to collaborate to address oral diseases with an emphasis in caries, accounting for structural determinants and implementing policies and actions that promote equitable access to quality dental services [4,5].

Often inequalities are attributed to individual socioeconomic and demographic factors. Although individual factors are important to explain health inequalities, the social structure that shapes them, including the social and political contexts of nations, should not be ignored [6]. The conceptual framework for action on the social determinants of health proposed by the World Health Organization (WHO) postulates that structural determinants generate or reinforce social stratification. Within this proposed framework, five key components constitute the structural, macro determinants: governance, macroeconomic policies, social policies, public policies, and cultural/societal values [7]. The societal context is integrate to the origin and maintenance of the distribution of power, prestige, and access to material resources, influencing the pattern of social stratification and class relations [7]. Therefore, it is possible to address the effects of the structural determinants of health inequalities through targeted actions on contextual characteristics, particularly the political dimension. Using social epidemiology methods and analytical tools to examine the structural determinants is still recent, even more so in oral health research [4,8].

Previous studies highlighted that, although oral health problems are experienced individually, they have important public and political characters [4,8]. The literature

suggests that oral health is related to social policies for income equality and economic measures, such as Gross Domestic Product (GDP) [8]. Social welfare regimes are also associated with better availability of resources, health education, and quality of life related to oral health [9,10]. Although prior research examined this issue to some extent, an understanding of the gradients of inequality between countries from a global perspective can provide further insights about the relationship between oral health and social and economic policies [11]. It may also provide valuable information about the availability of resources and disparities between countries and regions, through an understanding of the distribution of disease by structural determinants. This is also crucial in the identification of the best practices and policies to improve global oral health [8,12]. Assessing the distribution of untreated dental caries from the perspective of structural determinants expands the understanding of how governance, economic policies, and social contexts shape inequalities, contributing to inform of policies aimed at reducing disparities in oral health.

Based on the information presented above this study aims to carry out a comprehensive global assessment of the distribution of untreated cavities among both sexes in 204 countries and territories over the years. 2000, 2010 and 2019, focusing on the dimensions of structural determinants of health, on the conceptual framework proposed by the WHO. The study seeks to provide a deeper understanding of the distribution of untreated caries prevalence, incidence and years lived with disability (YLDs) in both primary and permanent dentitions by structural determinants of health that include dimensions of governance, macroeconomic policy, social policy, pub public policies and cultural/social values.

## Methods

### Study design

This descriptive study reported the incidence, prevalence, and years lived with disability (YLDs) due to untreated dental caries in deciduous and permanent teeth in 204 countries and territories by structural determinants of health, in specific, governance, macroeconomic policy, social policy, public policies and culture/societal values. The estimates of untreated caries in deciduous teeth were reported for children under 5 years of age, whereas estimates of untreated caries in permanent teeth were reported for adolescents and adults between 15 and 49 years of age. All estimates were reported for the years 2000, 2010, and 2019. Estimates of caries were retrieved from the Global Health Data Exchange (http://ghdx.healthdata.org/gbd-results-tool). Data collection and analysis can be divided into specific steps in addition, the full case definitions and inclusion/exclusion criteria for the literature review and measurement of global data on untreated caries are presented and described in the previous literature [2,11,13]. This study followed the Guidelines for Accurate and Transparent Health Estimates Reporting (GATHER) statement to ensure transparent reporting of the results (Supplementary material).

### Case definitions

Untreated caries in permanent teeth is defined as permanent dentition showing unmistakable cavity, undermined enamel, a detectably softened floor or wall, a tooth with a temporary filling or a tooth that is filled but also decayed [14]. Similarly, untreated caries of deciduous teeth was defined as deciduous dentition with an unmistakable cavity, undermined enamel, a detectably softened floor or wall, a tooth with a temporary filling or a tooth that is filled but also decayed, or teeth extracted due to caries [14].

### Metrics

The selection of structural determinants was strongly based on "A conceptual framework for action on the social determinants of health," considering the socioeconomic and political context block and the five proposed dimensions:

### Macroeconomic policy

The macroeconomic policy metric used was the gross domestic product per capita, popularly known as GDP per capita, which is an estimate of the total value of goods and services produced in a country during a given period. It is calculated

annually to provide a metric of the country's growth [15] (available at: https://microdata.worldbank.org/index.php/home). The countries' GDP data were collected from the World Development Indicators (WDI). Countries were categorized into "low", "medium" or "high" GDP, as this categorization was frequently used in previous studies [8].

## Public policy

Health expenditure was used as an indicator of public policies, which includes direct expenditures aimed at improving the health status of the population and/or distributing medical goods and services to the population [16]. The Global Health Expenditure Database (GHED) provides comparable data on health expenditure for more than 190 WHO Member States since 2000 with open access to the public [8]. Health spending as a proportion of government spending was categorized into terciles, to facilitate the interpretation of health inequalities. (available at: https://data.worldbank.org/indicator/SH.XPD.PUBL.GX.ZS?view=chart) [8].

## Governance

Six dimensions of governance are evaluated for 204 geographies using 32 individual data sources from various research institutes, international organizations, non-governmental agencies, and private sector companies, as detailed in 0 [17]. These dimensions include: Voice and Accountability, Political Stability and Absence of Violence/Terrorism, Government Effectiveness, Regulatory Quality, Rule of Law, and Control of Corruption. The sources used for each dimension are thoroughly explained in the original reference, providing a comprehensive methodology for the assessment, [17] as follows:

1º- Voice and accountability: perceptions of the degree to which a country's citizens are able to participate in choosing their government, as well as freedom of expression, freedom of association and free media;

2°- Political stability and absence of violence/terrorism: perceptions of the likelihood of political instability and/or politically motivated violence;

3º - Government effectiveness: perceptions of the quality of public services, civil service and the degree of its independence from political pressures, quality of policy formulation and implementation, and credibility of the government's commitment to policy;

4° - Quality of the regulator: perception of the government's ability to formulate and implement policy, and strong regulations that allow and promote the development of the private sector;

5° - Rule of law: perception of the degree to which agents trust and comply with the rules of society and, in particular, the quality of the execution of contracts, property rights, police and the courts, as well as the likelihood of crime and violence;

6º - Control of corruption: perception of the extent to which public power is exercised for private gain, including minor and major forms of corruption, as well as state "capture" by elites and private interests.

The percentage ranking across all countries is then calculated, ranging between 0 (lowest) and 1000 (highest), with countries categorized into terciles as "high", "medium" or "low" in terms of governance [8].

## Social policy

The sociodemographic index (SDI) was the indicator used to measure social policy. The SDI is an attempt to include more information to characterize a country's level of development, rather than relying on a dichotomous econometric measure or descriptor such as "developed/developing" [18,13]. Expressed on a scale ranging between 0 and 1, SDI is a composite average of the rankings of the incomes per capita, average educational attainment, and fertility rates of all countries and locations in the GBD study. Countries are then organized into SDI Qes [18,13].

## Culture/Societal values

The percentage of women holding seats in national parliaments is determined by calculating the number of seats occupied by women members in the single or lower chambers of these parliaments and expressing it as a percentage of the total number of occupied seats. This calculation involves dividing the total number of seats occupied by women by the overall number of seats in parliament. It should be noted that national parliaments can have either a bicameral or unicameral structure. In the case of unicameral parliaments, this indicator pertains to the single chamber, while in bicameral parliaments, it pertains to the lower chamber. The upper chamber of bicameral parliaments is not included in this calculation. The acquisition of seats typically occurs through general parliamentary elections, although nomination, appointment, indirect election, rotation of members, and by-elections can also be means of filling seats. The term "seats" refers to the number of parliamentary mandates or the number of members of parliament [19,20]. Proportion of seats held by women in national parliaments (%). This variable was categorized into tertiles to facilitate the understanding of the data distribution. The percentage of women in national parliaments is considered a relevant measure of societal values, particularly in terms of gender equality and political inclusion, which are central aspects of cultural and societal structures in a given country [19,20] (available at: https://data.worldbank.org/indicator/SG.GEN.PARL.ZS).

## Statement of Ethics

This study exclusively used secondary data publicly available from the Global Burden of Disease Study and the World Bank. As these data are aggregated and do not identify individuals, the research does not pose risks to privacy or confidentiality. Therefore, this study is exempt from approval by an Ethics Committee, in accordance with international ethical guidelines for studies based on secondary data.

## Statistical analysis

To obtain internally consistent estimates of prevalence, incidence and YLDs, the GBD uses DisMod-MR 2.1, a Bayesian meta-regression framework developed for GBD 2010 [2,13]. DisMod-MR 2.1 consolidated the code into a single language, Python, making it more transparent, more computationally efficient and better capable of ensuring internal consistency at the subnational level, even with limited data [11,14].

The descriptive analysis includes both sexes and examines: 1) untreated caries in primary teeth in children under 5 years of age and 2) untreated caries in permanent teeth in adolescents and adults between 15 and 49 years of age for the years 2000, 2010 and 2019. These years were chosen due to the availability of comparable data for all structural determinants included in the study. In addition, these periods comprise two decades, allowing the visualization of the distribution of caries at two distinct moments. This period also allows the identification of substantial patterns in the global burden due to caries and public health policy, social policy, governance, macroeconomic conditions and societal values.

Estimates of prevalence, incidence and YLDs (cases/100,000 people) [13,21]. Years Lived with Disability (YLDs) estimate of the number of years lost due to disability caused by a specific disease or health condition [14,18]. According to the GBD definition, the disability associated with untreated symptomatic caries is defined as "toothache, causing some difficulty eating" [22]. The methodology for calculating YLDs involves multiplying the prevalence of the health condition in a given population by its disability weight, which is measured on a scale from 0 to 1. The result is then multiplied by the time lived with the condition, in order to produce a measure of years lived with disability [14,18]. We present in Fig 1 a conceptual framework that combines structural determinants and untreated caries and highlight the interactive processes, drawing heavily on the World Health Organization's Conceptual Framework for Action on Social Determinants of Health [7].

Descriptive results were produced using Stata 14, and reported as rates or percentages, along with their 95% confidence intervals (CI).

**Fig 1. Conceptual theoretical model representing structural determinants of untreated dental caries on a World Health Organization Conceptual Framework for Action on Social Determinants of Health.**

## Results

The global distribution of untreated dental caries of deciduous teeth by structural determinants is presented in Tables 1 and 2, while Tables 3 and 4 present the distribution of untreated dental caries of permanent teeth by structural determinants. All rates were expressed per 100,000 people (rate).

The results of the analysis of untreated caries in deciduous teeth revealed a social gradient, with more developed countries showing smaller prevalence, incidence and YLDs. Table 1 shows the prevalence of untreated caries in primary dentition by structural determinants of health in 2000, 2010 and 2019. In 2019, high GDP countries had a mean untreated deciduous dental caries prevalence of 32,373.3/100,000 (95% CI [29,561.5−35,185.2]), whereas middle and low GDP countries had mean prevalence of 40,587.2/100,000 (95% IC [39,514.5−41,660.0]) and 36,573.1/100,000 (95% CI [35,313.0−37833.1]), respectively. In 2019, differences were observed in the prevalence of health expenditures, with countries in the lower and middle tertiles presenting the highest mean prevalence of untreated caries, 38,384.6/100,000 (95% CI [36,814.8−39,954.4]) and 37,204.0 (95% CI [35,792.3−38,615.6]), respectively. The upper tertile had a mean prevalence of 32,755.7 (95% CI [30,145.0−35,366.5]). Regarding the Sociodemographic index (SDI), the third quintile presented the highest mean prevalence, 39,319.1/100,000 (95% CI [37,903.3−40,734.9]), while the upper quintile presented the smallest mean prevalence, 30,289.9/100,000. (95% CI [26550.2−34029.5]). In respect to the variable 'women with seats in national parliaments', countries where women occupied more seats had lower mean deciduous caries prevalence, at 33,233.0/100,000 (95% CI [30,749.2−35,716.9])], when compared to countries with fewer women with seats in parliament, with a mean prevalence of 37,926.5/100,000 (95% CI [36,776.5−39,076.5]). A similar pattern is visible in 2000, 2010 and 2019 for the incidence and YLDs. For most structural determinants, YLDs increased over time. The social gradient was spanned over the two-decades period analyzed (Tables 1 and 2).

The findings regarding untreated dental caries in permanent teeth revealed differences in the mean prevalence across different GDP categories in 2000 (Table 3). Low GDP countries had a mean prevalence of 37,954.6 (95% CI [37,096.7−38,812.5]) whereas high income countries had a mean prevalence of 34,715.8 (95% CI [33,167.2−36,264.3]). In 2010, countries in the lower GDP category had a mean prevalence of 37,366.2/100,000 (95% CI [36,546.1−38,186.3]), whereas those in the upper category had a mean prevalence of 34,634.4/100,000 (95% CI [33488.8−35779.9]). The data show minimal variation in the mean untreated dental caries prevalence across tertiles of health expenditure. In the lowest tertile (T1), prevalences were 36,036.9 in 2000, 36,654.4 in 2010, and 35,898.2 in 2019, while in the highest tertile (T3), values

**Table 1. Prevalence and incidence of untreated caries in primary dentition by structural determinants of health in the years 2000, 2010 and 2019.**

| Structural determinants | Prevalence (2000) | | Prevalence (2010) | | Prevalence (2019) | | Incidence (2000) | | Incidence (2010) | | Incidence (2019) | |
|---|---|---|---|---|---|---|---|---|---|---|---|---|
| | Mean | IC95% | Mean | IC95% | Mean | IC95% | Mean | IC95% | Mean | IC95% | Mean | IC95% |
| Gross Domestic Product (GDP) | | | | | | | | | | | | |
| Low | 34708.0 | 32859.7 - 36556.2 | 37349.3 | 36222.2 - 38476.3 | 36573.1 | 35313.0 - 37833.1 | 42519.4 | 40866.1 - 44172.6 | 46909.7 | 45788.2 - 48031.2 | 46361.2 | 45156.1 - 47566.3 |
| Medium | 33740.5 | 31966.1 - 35514.9 | 39822.1 | 11714.9 - 41070.8 | 40587.2 | 39514.5 - 41660.0 | 42773.0 | 41331.2 - 44214.8 | 50157.8 | 49238.3 - 51077.3 | 50738.6 | 49886.1 - 51591.1 |
| High | 27522.4 | 24475.2 - 30569.6 | 26876.2 | 23373.6 - 30378.8 | 32373.3 | 29561.5 - 35185.2 | 37087.7 | 33936.4 - 40238.9 | 39438.6 | 35489.1 - 43388.2 | 45305.3 | 42995.0 - 47615.6 |
| Health expenditure (tertile) | | | | | | | | | | | | |
| T1 | 35049.4 | 33833.9– 36264.9 | 36770.4 | 35424.5 - 38116.2 | 37204.0 | 35792.3 - 38615.6 | 43489.3 | 42480.3 - 44498.3 | 46275.3 | 44977.3 - 47573.3 | 47136.3 | 45990.0 - 48282.6 |
| T2 | 29865.4 | 25442.6– 34288.2 | 37204.6 | 35293.0 - 39116.1 | 38384.6 | 36814.8 - 39954.4 | 38626.9 | 34004.9 - 43248.9 | 48101.5 | 46511.3 - 49691.7 | 48912.6 | 47811.4 - 50013.8 |
| T3 | 21597.3 | 18034.6– 25160.1 | 31208.8 | 28258.9 - 34158.6 | 32755.7 | 30145.0 - 35366.5 | 31757.6 | 27554.2 - 35961.1 | 42906.9 | 39830.4 - 45383.4 | 45456.4 | 43322.4 - 47590.3 |
| Governance | | | | | | | | | | | | |
| Low | 34277.7 | 32500.8 - 36054.6 | 36954.7 | 35631.5 - 38278.0 | 36128.9 | 34620.5 - 37637.3 | 43075.2 | 41621.0 - 44529.4 | 46762.9 | 45458.7 - 48067.1 | 46815.8 | 45453.4 - 48178.1 |
| Medium | 33913.8 | 31567.4 - 36260.1 | 38409.2 | 37057.6 - 39760.9 | 38930.0 | 37795.2 - 40064.7 | 42415.8 | 40207.7 - 44623.9 | 48309.0 | 47063.3 - 49554.6 | 49634.9 | 48760.0 - 50509.7 |
| High | 27699.3 | 24768.7 - 30629.9 | 31671.4 | 28795.5 - 34547.3 | 31746.7 | 28776.0 - 34717.4 | 36930.7 | 34050.6 - 39810.8 | 43254.1 | 40317.0 - 46191.1 | 45657.5 | 43637.5 47677.5 |
| Sociodemographic index (SDI) (quintiles) | | | | | | | | | | | | |
| Q1 | 30526.9 | 28796.5 - 32257.3 | 35331.3 | 33730.1 - 36932.5 | 36540.6 | 35349.8 - 37731.4 | 39919.6 | 38275.4 - 41563.7 | 44652.4 | 42988.5 - 46316.3 | 46064.9 | 45163.7 - 46966.1 |
| Q2 | 33508.4 | 31507.8 - 35509.1 | 36369.0 | 34533.1 - 38204.8 | 37868.2 | 36235.8 - 39500.6 | 42177.6 | 40482.8 - 43872.3 | 46442.7 | 44687.8 - 48197.6 | 48273.2 | 46988.2 - 49558.2 |
| Q3 | 35693.1 | 33154.8 - 38231.4 | 38182.5 | 36508.4 - 39856.7 | 39319.1 | 37903.3 - 40734.9 | 44426.4 | 42216.8 - 46636.1 | 49444.0 | 48245.4 - 50642.6 | 50004.6 | 48923.5 - 51085.7 |
| Q4 | 35086.4 | 31835.9 - 38336.9 | 37686.8 | 34657.7 - 40715.8 | 38292.8 | 35993.7 - 40591.8 | 43017.2 | 39977.5 - 46057.0 | 48473.8 | 45824.7 - 51122.9 | 49180.6 | 47707.0 - 50654.1 |
| Q5 | 25165.4 | 21189.2 - 29141.5 | 29329.1 | 25192.8 - 33465.3 | 30289.9 | 26550.2 - 34029.5 | 34757.7 | 30708.6 - 38806.9 | 40776.6 | 36454.2 - 45099.1 | 43742.8 | 40617.5 - 46868.2 |
| Women holding seats in national parliaments (tertile) | | | | | | | | | | | | |
| T1 | 37636.5 | 36354.7 - 38918.3 | 37574.4 | 36277.4 - 38871.4 | 37926.5 | 36776.5 - 39076.5 | 47187.0 | 45933.8 - 48440.3 | 47456.8 | 46263.4 - 48650.3 | 47422.02 | 46204.5 - 48639.5 |
| T2 | 37493.4 | 35471.4 - 39515.5 | 35966.4 | 33799.6 - 38133.1 | 37667.4 | 35901.6 - 39433.3 | 47830.8 | 46168.6 - 49493.0 | 47346.6 | 45448.2 - 49245.0 | 46793.63 | 45018.8 - 48568.3 |
| T3 | 33200.7 | 30749.7 - 35651.6 | 32176.0 | 29382.7 - 34969.4 | 33233.0 | 30749.2 - 35716.9 | 44258.8 | 42026.4 - 46491.3 | 43047.5 | 40272.0 - 45822.9 | 43397.75 | 40547.0 - 46248.4 |

95%CI: 95% of confidence interval.

*For both sexes and children under 5 years old.

All data were expressed as per 100,000 populations (rate).

**Table 2. Years Lived with Disability (YLDs) due to untreated caries in primary dentition by structural determinants of health in 2000, 2010 and 2019.**

| Structural determinants | YLD (2000) | | YLD (2010) | | YLD (2019) | |
|---|---|---|---|---|---|---|
| | | | Mean | IC95% | Mean | IC95% |
| Gross Domestic Product (GDP) | | | | | | |
| Low | 13.2 | 12.5–13.9 | 14.2 | 13.8 - 14.6 | 13.9 | 13.4 - 14.4 |
| Medium | 12.8 | 12.2–13.5 | 15.2 | 14.7 - 15.7 | 15.5 | 15.1 - 15.9 |
| High | 10.5 | 9.3–11.7 | 10.3 | 8.9 - 11.6 | 12.4 | 11.3 - 13.4 |
| Health expenditure (tertile) | | | | | | |
| T1 | 13.3 | 12.9–13.8 | 14.0 | 13.5 - 14.5 | 14.2 | 13.6 - 14.7 |
| T2 | 11.4 | 9.7–13.1 | 14.2 | 13.5 - 14.9 | 14.6 | 14.0 - 15.2 |
| T3 | 8.2 | 6.9–9.6 | 11.9 | 10.8 - 13.0 | 12.5 | 11.5 - 13.5 |
| Governance | | | | | | |
| Low | 13.0 | 12.4–13.7 | 14.1 | 13.6 - 14.6 | 13.7 | 13.2 - 14.3 |
| Medium | 12.9 | 12.0–13.8 | 14.6 | 14.1 - 15.2 | 14.8 | 14.4 - 15.3 |
| High | 10.6 | 9.4–11.7 | 12.1 | 11.0 - 13.2 | 12.1 | 11.0 - 13.2 |
| Sociodemographic index (SDI) (quintiles) | | | | | | |
| Q1 | 11.6 | 10.9–12.2 | 13.4 | 12.8 - 14.0 | 13.9 | 13.4 - 14.4 |
| Q2 | 12.8 | 12.0–13.5 | 13.8 | 13.1 - 14.5 | 14.4 | 13.8 - 15.0 |
| Q3 | 13.6 | 12.6–14.6 | 14.6 | 13.9 - 15.2 | 15.0 | 14.5 - 15.5 |
| Q4 | 13.4 | 12.2–14.6 | 14.4 | 13.2 - 15.5 | 14.6 | 13.7 - 15.5 |
| Q5 | 9.6 | 8.1–1.1 | 11.2 | 9.6 - 12.8 | 11.6 | 10.1 - 13.0 |
| Women holding seats in national parliaments (tertile) | | | | | | |
| T1 | 14.3 | 13.8–14.8 | 14.3 | 13.8–14.8 | 14.2 | 13.8–14.7 |
| T2 | 14.3 | 13.5–15.1 | 13.7 | 12.9–14.5 | 13.8 | 13.0–14.6 |
| T3 | 12.6 | 11.7–13.6 | 12.3 | 11.2–13.3 | 12.3 | 11.2–13.3 |

95%CI: 95% of confidence interval.

*For both sexes and children under 5 years old.

All data were expressed as per 100,000 populations (rate).

were 36,491.6, 36,340.9, and 35,942.1, respectively. The differences were not substantial, suggesting stability in health expenditure. Regarding women's representation in parliament, the mean prevalence of untreated caries also showed little fluctuation: in T1, prevalences were 36,830.0 in 2000, 36,426.3 in 2010, and 36,000.7 in 2019; in T3, they were 36,688.1, 36,393.3, and 35,749.1, respectively. These results suggest that neither health expenditure nor female representation in parliament significantly influenced the mean prevalence and incidence of untreated caries over time. Between-country inequities were also observed in relation to governance and the socioeconomic index. However, no gradients were observed in respect to incidence or YLDs, irrespective of the year (Table 4). In addition to the results presented, supplementary country-level information is provided in S1 Appendix A, S2 Appendix B, and S3 Appendix C, including: in S1 Appendix A, the macrostructural determinants of health in 2000, 2010, and 2019; in S2 Appendix B, data on the prevalence, incidence, and years lived with disability (YLDs) due to caries in the primary dentition; and in S3 Appendix C, the same indicators related to permanent dentition, all referring to the same years.

## Discussion

The findings revealed a possible between-country social gradients in untreated caries, for both deciduous and permanent teeth. These gradients were more pronounced for untreated caries in deciduous teeth, and persisted from 2000, 2010–2019, meaning that unfair differences between poor and rich countries in untreated caries did not change, regardless

**Table 3. Prevalence and incidence of untreated caries in permanent dentition by structural determinants of health in 2000, 2010 and 2019.**

| Structural determinants | Prevalence (2000) | | Prevalence (2010) | | Prevalence (2019) | | Incidence (2000) | | Incidence (2010) | | Incidence (2019) | |
|---|---|---|---|---|---|---|---|---|---|---|---|---|
| | Mean | IC95% | Mean | IC95% | Mean | IC95% | Mean | IC95% | Mean | IC95% | Mean | IC95% |
| Gross Domestic Product (GDP) | | | | | | | | | | | | |
| Low | 37954.6 | 37096.7 - 38812.5 | 37366.2 | 36546.1 - 38186.3 | 36592.8 | 35639.7 - 37546.0 | 48848.5 | 48326.2 - 49370.7 | 49471.0 | 48979.6– 49962.4 | 49880.3 | 49339.3– 50421.3 |
| Medium | 35432.4 | 33621.9 - 37242.9 | 36878.0 | 34886.4 - 38869.5 | 36485.8 | 34867.9 - 38103.7 | 49215.8 | 48170.9 - 50260.6 | 48346.2 | 47229.1– 49463.4 | 48892.7 | 48051.8– 49733.6 |
| High | 34715.8 | 33167.2 - 36264.3 | 34634.4 | 33488.8 - 35779.9 | 35229.2 | 34064.8 - 36393.6 | 50141.5 | 49507.4 - 50775.6 | 50203.9 | 49722.3– 50685.5 | 49480.4 | 48882.7– 50078.2 |
| Health expenditure (tertile) | | | | | | | | | | | | |
| T1 | 36036.9 | 34806.5 - 37267.3 | 36654.4 | 35582.3 - 37726.4 | 35898.2 | 34878.8 - 36917.5 | 50036.7 | 49320.1 - 50753.4 | 50157.7 | 49519.5– 50795.9 | 50273.4 | 49654.1– 50892.7 |
| T2 | 38677.7 | 37313.6 - 40041.7 | 36836.6 | 35636.7 - 38036.5 | 36422.6 | 35095.0 - 37750.2 | 48167.3 | 47365.5 - 48969.2 | 49105.3 | 48516.7– 49694.0 | 49161.0 | 48534.2– 49787.9 |
| T3 | 36491.6 | 35284.7 - 37698.6 | 36340.9 | 34783.7 - 37898.2 | 35942.1 | 34686.4 - 37197.8 | 49215.6 | 48550.8 - 49880.4 | 48951.0 | 48055.9– 49846.1 | 49205.9 | 48530.6– 49881.2 |
| Governance | | | | | | | | | | | | |
| Low | 37731.3 | 36549.9 - 38912.7 | 37011.0 | 35964.6 - 38057.3 | 36925.5 | 35943.0 - 37908.0 | 49116.5 | 48382.2 - 49850.9 | 49581.0 | 48951.8– 50210.1 | 49703.2 | 49131.4– 50275.0 |
| Medium | 37744.8 | 36316.4 - 39173.2 | 37567.8 | 36153.2 - 38982.4 | 37471.4 | 35978.0 - 38964.9 | 48703.0 | 47877.3 - 49528.7 | 48965.2 | 48121.9– 49808.6 | 48790.8 | 47909.8– 49671.8 |
| High | 35592.5 | 34416.0 - 36769.0 | 34967.2 | 33849.0 - 36085.4 | 35342.1 | 34177.2 - 36506.9 | 49509.0 | 48891.2 - 50126.9 | 49623.4 | 49091.4– 50155.3 | 49580.6 | 49011.5– 50149.8 |
| Sociodemographic index (SDI) (quintiles) | | | | | | | | | | | | |
| Q1 | 37626.5 | 36297.4 - 38955.6 | 37401.3 | 36282.3 - 38520.3 | 36417.4 | 35167.2 - 37667.6 | 49672.6 | 49050.5 - 50294.8 | 49902.9 | 49425.4– 50380.5 | 50333.9 | 49692.9– 50974.8 |
| Q2 | 36557.7 | 35407.6 - 37707.8 | 35576.8 | 34082.6 - 37071.0 | 36352.4 | 34903.8 - 37801.0 | 49669.9 | 48938.2 - 50401.5 | 50127.8 | 49202.6– 51053.0 | 49421.1 | 48608.9– 50233.4 |
| Q3 | 37540.3 | 35611.4 - 39469.2 | 35978.6 | 34436.3 - 37520.9 | 36099.2 | 34402.0 - 37796.4 | 48782.4 | 47662.5 - 49902.3 | 49347.3 | 48585.5– 50109.1 | 49328.7 | 48468.9– 50188.5 |
| Q4 | 38645.9 | 36783.3 - 40508.4 | 38582.2 | 36517.9 - 40646.5 | 38292.8 | 35285.3 - 38705.9 | 47460.1 | 46402.1 - 48518.1 | 47693.9 | 46462.5– 48925.2 | 48437.9 | 47524.7– 49351.1 |
| Q5 | 34124.7 | 32791.7 - 35457.6 | 34398.3 | 32936.4 - 35860.2 | 30289.9 | 33224.1 - 35843.2 | 50233.5 | 49602.4 - 50864.7 | 50016.8 | 49358.2– 50675.5 | 49961.0 | 49294.0– 50628.0 |
| Women holding seats in national parliaments (tertile) | | | | | | | | | | | | |
| T1 | 36830.0 | 35798.2 - 37861.8 | 36426.3 | 35385.7 - 37466.8 | 36000.7 | 34987.6 - 37013.8 | 49510.0 | 48987.2 - 50032.8 | 49763.6 | 49267.1 - 50260.2 | 49836.9 | 49403.2 - 50270.6 |
| T2 | 37840.2 | 36440.1 - 39240.4 | 37169.6 | 35910.6 - 38428.5 | 36824.6 | 35537.5 - 38111.6 | 48500.6 | 47628.9 - 49372.3 | 48914.3 | 48181.1 - 49647.5 | 49191.0 | 48411.0 - 49971.0 |
| T3 | 36688.1 | 35359.7 - 38016.5 | 36393.28 | 34880.5 - 37906.0 | 35749.1 | 34573.4 - 36924.9 | 49354.2 | 48627.4 - 50081.0 | 49335.6 | 48462.4 - 50208.7 | 49430.4 | 48803.5 - 50057.3 |

95%CI: 95% of confidence interval.

*For both sexes and children under 5 years old.

All data were expressed as per 100,000 populations (rate).

**Table 4. Years Lived with Disability (YLDs) due to untreated caries in permanent dentition by structural determinants of health in 2000, 2010 and 2019.**

| Structural determinants | YLD (2000) | | YLDs (2010) | | YLDs (2019) | |
|---|---|---|---|---|---|---|
| | Mean | | Mean | IC95% | Mean | IC95% |
| Gross Domestic Product (GDP) | | | | | | |
| Low | 37.6 | 36.8 - 38.5 | 37.1 | 36.3 - 37.9 | 36.3 | 35.4 - 37.2 |
| Medium | 35.2 | 33.4 - 37.0 | 36.6 | 34.6 - 38.6 | 36.2 | 34.6 - 37.8 |
| High | 34.4 | 32.9 - 36.0 | 34.3 | 33.2 - 35.5 | 34.9 | 33.8 - 36.1 |
| Health expenditure (tertile) | | | | | | |
| T1 | 35.7 | 34.5 - 37.0 | 36.4 | 35.3 - 37.4 | 35.6 | 34.6 - 36.6 |
| T2 | 38.4 | 37.0 - 39.7 | 36.6 | 35.4 - 37.8 | 36.1 | 34.8 - 37.4 |
| T3 | 36.2 | 35.0 - 37.4 | 36.0 | 34.5 - 37.6 | 35.6 | 34.4 - 36.9 |
| Governance | | | | | | |
| Low | 37.4 | 36.2 - 38.6 | 36.7 | 35.7 - 37.8 | 36.6 | 35.7 - 37.6 |
| Medium | 37.5 | 36.0 - 38.9 | 37.3 | 35.9 - 38.7 | 37.2 | 35.7 - 38.7 |
| High | 35.3 | 34.1 - 36.5 | 34.7 | 33.6 - 35.8 | 35.1 | 33.9 - 36.2 |
| Sociodemographic index (SDI) (quintiles) | | | | | | |
| Q1 | 37.2 | 35.9 - 38.5 | 37.1 | 36.0 - 38.2 | 36.1 | 34.9 - 37.3 |
| Q2 | 36.3 | 35.1 - 37.4 | 35.3 | 33.8 - 36.8 | 36.1 | 34.6 - 37.5 |
| Q3 | 37.3 | 35.4 - 39.2 | 35.7 | 34.2 - 37.3 | 35.8 | 34.1 - 37.5 |
| Q4 | 38.4 | 36.5 - 40.2 | 38.3 | 36.3 - 40.4 | 36.7 | 35.0 - 38.4 |
| Q5 | 33.8 | 32.5 - 35.2 | 34.1 | 32.7 - 35.6 | 34.2 | 32.9 - 35.5 |
| Women holding seats in national parliaments (tertile) | | | | | | |
| T1 | 36.5 | 35.5–37.6 | 36.2 | 35.1–37.2 | 35.7 | 34.7–36.7 |
| T2 | 37.5 | 36.2–38.9 | 36.9 | 35.6–38.1 | 36.5 | 35.3–37.8 |
| T3 | 36.4 | 35.0–37.7 | 36.1 | 34.6–37.6 | 35.4 | 34.3–36.6 |

95%CI: 95% of confidence interval.

*For both sexes and children under 5 years old.

All data were expressed as per 100,000 populations (rate).

of an overall economic growth during this period. The social gradients can be explained by the observed living conditions and positive changes in social and public policies in some regions. However, disparities in access to oral health services and differences in availability and access to fluoride may also contribute to these differences [5,23]. While the literature on inequalities within countries is more common, the results highlight the importance of differences between countries, which can help identify public oral health policies that need to be reviewed and reflect on tackling inequities in oral health globally.

Some limitations of this study need to be accounted for. While the GBD uses secondary data to produce global estimates of disease burden, there are challenges in reviewing dental literature for reporting untreated caries. Most studies report caries experience as the average number of decayed, missing, or filled teeth, and the relationship between untreated caries (DT>0) and lifetime prevalence (DMFT>0) is not constant and has not been quantified [24]. As a result, countries with only DMFT>0 data are excluded because DisMod-MR relies on fixed effects being relatively constant over time and age [24,25]. The lack of data in some areas of the globe and the quality of published and unpublished data are also challenges. To address these issues, few studies using non-probabilistic sampling were included, but their impact on the results is limited because DisMod-MR adjusts caries prevalence estimates by a population-weighted average correction factor in a hierarchy of super-region, region, and country [21]. Another limitation is the measurement of cultural or societal values through a single proxy emphasizing gender equity. While gender equity has been used as proxy of culture [19,20], this is a complex social construct that encompasses beliefs, values, norms and practices shared by a group of

people. This means that using gender equity lacks specificity and leads to measurement error that needs to be accounted for when interpreting the results. Another limitation is related to the use of government health expenditure, which is also prone to error, since expenditures in oral health are may not be correlated to expenses in general health. Furthermore, it is important to note that the age range was determined based on the availability of comparable data, although the difference between 5 and 15 years, particularly given the common focus on 6- and 12-year-olds in caries incidence studies, may affect comparing the findings of the present study with those of oral epidemiology surveys that used the ages recommended by the WHO pathfinder oral health surveys method. Despite these limitations, the unprecedented results of this study highlight consistent disparities in dental caries between countries over a two-decades period. These persistent inequalities underscore the urgent need for comprehensive approaches that align with the World Health Organization's recommendations to address them [3].

The distribution of dental caries in deciduous teeth varied across countries by the sociodemographic index, governance, and health expenditures. These findings align with the existing literature, suggesting that developing countries experience higher caries prevalence [8,26]. Governance and health expenditures are also barriers to oral health care, affecting the distribution of untreated caries. Better child oral health is linked to higher per capita GDP, greater social generosity, education, sociodemographic index, and lower income inequality [8,26]. Middle-income countries showed higher caries prevalence and years lived with disability (YLDs) when compared to lower-income countries. One possible explanation is that middle-income countries are experiencing a rapid nutrition transition, which is represented by a shift from traditional to Western-style, more energy-dense diets that are high in added sugars and associated with obesity and chronic diseases [27,28]. In contrast, low-income countries may have lower caries prevalence because traditional diets are the norm and ultra-processed foods high in added sugars, such as sugar-sweetened beverages, are less frequently consumed. In the specific case of low-income countries, limited access to dental care means that the majority of caries will go untreated [8,26]. Another possible explanation for the higher YLDs observed in primary teeth, compared to permanent teeth, is the lower likelihood of restorative treatment in the primary dentition. In many settings, primary teeth are perceived as temporary and less important, which may lead to under-treatment. For instance, a study conducted among parents in Chennai, India, found that 68% believed treatment of primary teeth was unnecessary because they would eventually shed, and 43% thought that missing spaces in primary dentition could be left unaddressed until the eruption of permanent teeth [29]. This perception contributes to reduced clinical attention and prioritization, resulting in a greater burden of untreated caries and, consequently, higher disability estimates in early childhood. These patterns may also reflect structural barriers to timely care, particularly among children at higher risk of caries.

The findings of this study reinforce the urgency of rethinking health systems, emphasizing universal health coverage and access to oral health services. Tooth decay is the most prevalent disease globally, affecting billions of people and generating significant economic impacts, whether through dental expenses (direct costs) or productivity losses (indirect costs) [30]. From an economic point of view, improvements in the population's oral health can be highly beneficial and contribute significantly to well-being. However, in many countries, health systems remain fragmented and strictly oriented towards disease treatment, reflecting inefficient and exclusionary health care models [30]. This limits access to oral health services, perpetuating inequalities and leading to uneven disease distribution. Low- and Middle-income countries also suffer more from the influence of large corporations and commercial drivers, contributing to the global increase in the consumption of foods high in sugar [31,32]. Additionally, food insecurity, which is more prevalent in these countries, often leads to a greater reliance on sugar-based foods due to their affordability, further exacerbating tooth decay and oral health disparities [33]. These dynamics may partially explain why countries in the third quintile of the SDI, despite having more resources than those in the lowest quintile, still have higher prevalence of caries [34]. This suggests that factors such as rapid nutrition transitions within the middle-income category [34]. This aspect emphasizes the need for an intersectoral approach based on a health promotion perspective to these countries [34]. This aspect emphasizes the need for a broader approach to oral health policies in low- and middle-income countries [34].

A collaborative approach that considers non-communicable diseases in an integrated manner seems to be a rational and more efficient action than approaches focused exclusively on oral disorders [35]. Such an approach also has the potential to reduce mortality rates attributed to cardiovascular diseases, cancer, and diabetes. Although individualized actions are commonly carried out and targeted at school populations, they are not effective in promoting oral health of the communities at large. They are also prone to duplicating health messages and being ineffective throughout life [35].

The Global Oral Health Action Plan 2023–2030, formulated under the auspices of the World Health Organization, presents six strategic objectives (Governance; Promotion and Prevention of Oral Health; Health Workforce; Oral Health Care; Information Systems; and Research) and 11 related targets to be achieved by 2030 [36]. Two of these targets are ambitious: by 2030, having 80% of the global population entitled to essential oral health care services, ensuring progress towards Universal Health Coverage (UHC) for oral health; and achieving a 10% relative reduction in the global lifetime prevalence of major oral diseases. While these goals may be difficult to achieve, they represent a positive political momentum for Global Oral Health, encouraging countries to implement oral health policies and programs aimed at tackling oral health inequalities, and fostering collaboration with the broader public health agenda and intersectoral work [36]. Achieving these goals require bold actions, including addressing gaps in global oral epidemiology and health information systems, and improving surveys and community oral health surveillance to provide timely, relevant information and current data for analysis; rigorous evaluation of evidence to promote equity in the prevention and treatment of oral diseases globally. Therefore, global estimates and more in depth analysis of estimates such as the ones produced by the GBD represent an opportunity to monitor progress towards the actions that are part of the WHO Global Oral Health Strategy [37].

In conclusion, this study provides a global assessment of untreated dental caries across 204 countries, revealing persistent inequalities linked to structural determinants of health. The findings indicate that countries with higher GDP and better sociodemographic indices generally show lower prevalence and YLDs due to untreated caries, particularly in deciduous teeth. In contrast, lower-income countries experience greater burden due to untreated caries, suggesting that policies should focus on improving access to dental care and addressing socio-economic disparities. Future research could further investigate the impact of oral health policies and socio-cultural factors on these global trends.

## Supporting information

**S1 Appendix A. Macro determinants of health by country in 2000, 2010 and 2019.**
(DOCX)

**S2 Appendix B. Prevalence, incidence and YLDs of caries in the primary dentition in 2000, 2010 and 2019 by country.**
(DOCX)

**S3 Appendix C. Prevalence, incidence and YLDs of caries in permanent dentition in 2000, 2010 and 2019 by country.**
(DOCX)

## Author contributions

**Conceptualization:** Orlando Luiz do Amaral Júnior, Maria Laura Braccini Fagundes, Fernando Neves Hugo, Nicholas J Kassebaum, Jessye Melgarejo do Amaral Giordani.

**Data curation:** Orlando Luiz do Amaral Júnior, Fernando Neves Hugo.

**Formal analysis:** Fernando Neves Hugo.

**Investigation:** Orlando Luiz do Amaral Júnior.

**Methodology:** Orlando Luiz do Amaral Júnior, Maria Laura Braccini Fagundes, Fernando Neves Hugo, Jessye Melgarejo do Amaral Giordani.

**Supervision:** Fernando Neves Hugo, Nicholas J Kassebaum, Jessye Melgarejo do Amaral Giordani.

**Validation:** Orlando Luiz do Amaral Júnior, Maria Laura Braccini Fagundes, Fernando Neves Hugo, Nicholas J Kassebaum, Jessye Melgarejo do Amaral Giordani.

**Visualization:** Orlando Luiz do Amaral Júnior, Maria Laura Braccini Fagundes, Fernando Neves Hugo, Nicholas J Kassebaum, Jessye Melgarejo do Amaral Giordani.

**Writing – original draft:** Orlando Luiz do Amaral Júnior, Maria Laura Braccini Fagundes, Fernando Neves Hugo, Nicholas J Kassebaum, Jessye Melgarejo do Amaral Giordani.

**Writing – review & editing:** Orlando Luiz do Amaral Júnior, Maria Laura Braccini Fagundes, Fernando Neves Hugo, Jessye Melgarejo do Amaral Giordani.

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
