## [Decision Letter · Decision Letter 0]

21 Mar 2025

PONE-D-25-05748Structural determinants of inequalities in untreated dental caries in the Global Burden of Disease StudyPLOS ONE

Dear Dr. Hugo,

Thank you for submitting your manuscript to PLOS ONE. After careful consideration, we feel that it has merit but requires some minor revisions to fully meet PLOS ONE’s publication criteria. Therefore, we invite you to submit a revised version of the manuscript that addresses the points raised during the review process.

We look forward to receiving your revised manuscript.

Kind regards,

Thiago Machado Ardenghi

Academic Editor

PLOS ONE

3. Please ensure that you refer to Figure 1 in your text as, if accepted, production will need this reference to link the reader to the figure.

Additional Editor Comments (if provided):

Reviewers' comments:

Reviewer's Responses to Questions

**Comments to the Author**

1. Is the manuscript technically sound, and do the data support the conclusions?

Reviewer #1: Yes

Reviewer #2: Yes

2. Has the statistical analysis been performed appropriately and rigorously? 

Reviewer #1: Yes

Reviewer #2: Yes

3. Have the authors made all data underlying the findings in their manuscript fully available?

Reviewer #1: Yes

Reviewer #2: Yes

4. Is the manuscript presented in an intelligible fashion and written in standard English?

Reviewer #1: Yes

Reviewer #2: Yes

5. Review Comments to the Author

Reviewer #1: The purpose of this investigation was to provide a comprehensive global assessment of the distribution of untreated dental caries in 204 countries and territories, spanning the years 2000, 2010, and 2019, by dimensions of structural determinants of health.

My overall judgment is that the paper addresses an important topic of public health, the manuscript is compatible with the journal, and it presents the necessary quality. The manuscript is well-written and detailed. It describes the importance of the study for public oral health, presents a robust and well-defined methodology, presents interesting results, and discusses relevant works. I recommend your acceptance.

- I suggest sending Figure 1 in high-definition.

Reviewer #2: Dear Authors,

I would like to congratulate you on your hard work in preparing the manuscript entitled

“Structural determinants of inequalities in untreated dental caries in the Global Burden of Disease Study”. It was a pleasure to review and learn from your study.

However, there are some points (listed below) that I believe should be addressed to enhance the manuscript quality.

Some of these suggestions are reflections that you may consider, or not, and incorporate into your work.

Sincerely,

INTRODUCTION:

- There are two dots (.) in line 46.

- Also in line 46: Wouldn’t “integrate” fit the sentence more than “integral”?

- Line 55: Maybe the authors are talking about social policies FOR income equality and economic measures?

- Through lines 57 to 60 the authors state, “Although prior research examined this issue to some extent, an understanding of the gradients of inequality between countries from a global perspective can provide further insights about the relationship between oral health and social and economic policies”. To what extent have the prior studies examined this issue? Why is your study different from these studies and why is it so relevant? This needs to be clearer in your introduction. What advances your study brings to science that the studies from Barnabe et al., 2017 (Global, Regional, and National Levels and Trends in Burden of Oral Conditions from 1990 to 2017: A Systematic Analysis for the Global Burden of Disease 2017 Study), Wen et al., 2022 (Global Burden and Inequality of Dental Caries, 1990 to 2019), Vos et al., 2020 (Global burden of 369 diseases and injuries in 204 countries and territories, 1990–2019: a systematic analysis for the Global Burden of Disease Study 2019) Zhao et al., 2023 (Burden, trends, and inequality of dental caries in the US, 1990–2019), and Qin et al., 2022 (A systematic analysis for the Global Burden of Disease study) have not yet addressed?

METHODS:

-Line 82: The authors say, “Data collection and analysis can be divided into specific steps”. What steps are those?

-The link provided in line 120 is no longer available.

-In line 126 the authors start describing how the governance variable was measured, and they explain the six dimensions evaluated. But where are the actual data sources to measure each one of these dimensions? The authors mention 32 individual data sources. I believe this should be explicitly illustrated. It would be helpful to know which data source was used for each dimension.

- It should be made clear that the 'Culture/Societal Values' structural determinant was measured by the percentage of women holding seats in national parliaments. However, why does this measure effectively represent this determinant? Why not consider other cultural and societal characteristics? This point needs to be addressed in the text.

- I understand that the authors chose the age range based on the availability of comparable data for all structural determinants. However, don't you think there is a significant and important gap between 5 and 15 years old? This is especially relevant because 6 and 12 years old are typically the ages at which many studies assess caries incidence. While I understand your reason, I believe it would be beneficial to acknowledge this limitation in the discussion section.

RESULTS:

-In line 216 I think it's important to reinforce that you are still talking about 2019 data to make it easier for the reader to find the number in the table. The same comment is valued for lines 218, 220, 221 and 223.

-In line 218 you say Regarding the Social Development Index (SDI), the third quintile presented the highest mean prevalence, 220 39,319.1/100,000 (95% CI [37,903.3 – 40,734.9]), while the upper quintile presented the smallest mean prevalence, 30,289.9/100,000. (95% CI [26550.2-34029.5]). The possible reason for this has to be in the discussion session.

-In line 219 you mention the Social Development Index (SDI) but in the table is written Sociodemographic index (SDI). Which one is correct?

-I think it's important that line 227 continues by noting that, for most structural determinants, YLDs increase over time. This point should be discussed further later in the manuscript.

-Why did the authors not provide data in the third paragraph (line 228) about health expenditure and Women holding seats in national parliaments for untreated caries in permanent dentition?

DISCUSSION:

- What is or are the possible explanation(s) for the incidence of between-country inequalities in YLDs not being observed in permanent teeth, but happening in primary teeth?

-Lines 263 and 264 are missing a reference.

-I agree when the authors say that “Low- and Middle-income countries also suffer more from the influence of large corporations and commercial drivers, contributing to the global increase in the consumption of foods high in sugar.26,27.” I also think it would be interesting to explore the aspect of food insecurity, which is more prevalent in these countries, leading to higher consumption of sugar-based foods because they are cheaper, leading to more dental caries.

-Why do the authors say nothing about YLDs in the conclusion? The conclusion must answer your aim.

6. PLOS authors have the option to publish the peer review history of their article (what does this mean? ). If published, this will include your full peer review and any attached files.

**Do you want your identity to be public for this peer review?** For information about this choice, including consent withdrawal, please see our Privacy Policy .

Reviewer #1: No

Reviewer #2: **Yes: ** Ana Luiza Peres Baldiotti, DDS, MSc, PhD

---

## [Author Response · Author response to Decision Letter 1]

9 Apr 2025

Review Comments to the Author

Reviewer #1: The purpose of this investigation was to provide a comprehensive global assessment of the distribution of untreated dental caries in 204 countries and territories, spanning the years 2000, 2010, and 2019, by dimensions of structural determinants of health.

My overall judgment is that the paper addresses an important topic of public health, the manuscript is compatible with the journal, and it presents the necessary quality. The manuscript is well-written and detailed. It describes the importance of the study for public oral health, presents a robust and well-defined methodology, presents interesting results, and discusses relevant works. I recommend your acceptance.

- I suggest sending Figure 1 in high-definition.

Response: Thank you for your careful review and positive evaluation of our manuscript. We appreciate your insightful comments and your recognition of the relevance and quality of our study. In response to your suggestion, we have resubmitted Figure 1 with improved resolution to ensure better clarity.

Reviewer #2: Dear Authors,

I would like to congratulate you on your hard work in preparing the manuscript entitled

“Structural determinants of inequalities in untreated dental caries in the Global Burden of Disease Study”. It was a pleasure to review and learn from your study.

However, there are some points (listed below) that I believe should be addressed to enhance the manuscript quality.

Some of these suggestions are reflections that you may consider, or not, and incorporate into your work.

Sincerely,

INTRODUCTION:

- There are two dots (.) in line 46.

- Also in line 46: Wouldn’t “integrate” fit the sentence more than “integral”?

- Line 55: Maybe the authors are talking about social policies FOR income equality and economic measures?

Response: Thank you for your suggestions. We have addressed all the requested changes accordingly.

- Through lines 57 to 60 the authors state, “Although prior research examined this issue to some extent, an understanding of the gradients of inequality between countries from a global perspective can provide further insights about the relationship between oral health and social and economic policies”. To what extent have the prior studies examined this issue? Why is your study different from these studies and why is it so relevant? This needs to be clearer in your introduction. What advances your study brings to science that the studies from Barnabe et al., 2017 (Global, Regional, and National Levels and Trends in Burden of Oral Conditions from 1990 to 2017: A Systematic Analysis for the Global Burden of Disease 2017 Study), Wen et al., 2022 (Global Burden and Inequality of Dental Caries, 1990 to 2019), Vos et al., 2020 (Global burden of 369 diseases and injuries in 204 countries and territories, 1990–2019: a systematic analysis for the Global Burden of Disease Study 2019) Zhao et al., 2023 (Burden, trends, and inequality of dental caries in the US, 1990–2019), and Qin et al., 2022 (A systematic analysis for the Global Burden of Disease study) have not yet addressed?

Response: Thank you for this comment. We fully appreciate the importance of the studies referenced, such as those by Bernabe et al. (2017), Wen et al. (2022), Vos et al. (2020), Zhao et al. (2023), and Qin et al. (2022), which have made valuable contributions to understanding the global burden of oral health conditions, including dental caries, and focusing on the distribution of oral disease burden by geography and Sociodemographic Index. The focus of the present study, however, is to provide an analysis of the distribution of untreated dental caries through the lens of structural determinants of health and using a robust theoretical framework to select variables capture those structural determinants, which the studies mentioned did not intend to do. This perspective emphasizes the role of governance, macroeconomic policies, social policies, and cultural values in shaping health inequalities, particularly in relation oral disease burden, thereby providing a more detailed and nuanced understanding of the contribution of these structural determinants.

In response to your suggestion, we have revised the third paragraph of the introduction to make this focus clearer. A sentence has been added at the end to highlight the unique contribution of this study. We hope this review addresses the presented concerns and further clarifies the relevance and distinctiveness of the study.

Revised text:

“Assessing the distribution of untreated dental caries from the perspective of structural determinants expands the understanding of how governance, economic policies, and social contexts shape inequalities, contributing to inform policies aimed at reducing oral health disparities.”

METHODS:

-Line 82: The authors say, “Data collection and analysis can be divided into specific steps”. What steps are those?

Response: Thank you for your feedback. When reviewing the text, we decided to remove the sentence “Data collection and analysis can be divided into specific steps,” as we agreed that it could cause some confusion for readers. Initially, we referred to the process of first collecting data of the Global Burden of Disease (GBD) Study from the Global Health Data Exchange repository, and then merging the World Bank dataset. By removing this sentence, we intended to make the description clearer and avoid any potential misunderstanding about the data analysis process.

-The link provided in line 120 is no longer available.

Response: Thank you for this important comment. We have re-inserted the updated link, so that post-revision the link to the World Bank Microdata Library is available.

-In line 126 the authors start describing how the governance variable was measured, and they explain the six dimensions evaluated. But where are the actual data sources to measure each one of these dimensions? The authors mention 32 individual data sources. I believe this should be explicitly illustrated. It would be helpful to know which data source was used for each dimension.

Response: Thank you for your comment. Based on your feedback, we have rewritten the paragraph to clarify the reference to governance dimensions and their data sources. We would like to highlight that the aim of our study was not to assess the impact of governance dimensions on oral health inequalities separately. However, full information on each source is available at Kaufmann et al. (2010).

The revised paragraph reads as follows:

Six dimensions of governance are evaluated for 204 geographies using 32 individual data sources from various research institutes, international organizations, non-governmental agencies, and private sector companies, as detailed in Kaufmann et al. (2010)17. These dimensions included: Voice and Accountability, Political Stability and Absence of Violence/Terrorism, Government Effectiveness, Regulatory Quality, Rule of Law, and Control of Corruption. The sources used for each dimension are thoroughly explained in the original reference, providing a comprehensive methodology for the assessment, 17 as follows:

- It should be made clear that the 'Culture/Societal Values' structural determinant was measured by the percentage of women holding seats in national parliaments. However, why does this measure effectively represent this determinant? Why not consider other cultural and societal characteristics? This point needs to be addressed in the text.

Response: Thank you for the suggestion. We agree that the choice of 'Culture/Social Values', measured by the percentage of women holding seats in national parliaments requires further explanation. This metric was selected because it might reflect social values related to gender equality and political inclusion, which are central to understanding the cultural and social determinants of health. This selection has its limitations, since gender equality does not fully captures culture, which is a multidimensional construct. However, we would also like to emphasize that, while other dimensions of culture may also be relevant, this metrics has been widely used in research and provides a tangible assessment of culture as a structural determinant of health inequalities. We have chosen to revise the text by including a sentence that makes these factors clearer.

Revised text:

The percentage of women in national parliaments is considered a relevant measure of societal values, particularly in terms of gender equality and political inclusion, which are central aspects of cultural and societal structures in a given society20,21.

References:

20. Mirziyoyeva Z, Salahodjaev R. Does representation of women in parliament promote economic growth? Considering evidence from Europe and Central Asia. Front Polit Sci. 2023;5:1120287. doi:10.3389/fpos.2023.1120287

21. Ng E, Muntaner C. The effect of women in government on population health: An ecological analysis among Canadian provinces, 1976–2009. SSM - Population Health. 2018;6:141-148. doi:10.1016/j.ssmph.2018.08.003

- I understand that the authors chose the age range based on the availability of comparable data for all structural determinants. However, don't you think there is a significant and important gap between 5 and 15 years old? This is especially relevant because 6 and 12 years old are typically the ages at which many studies assess caries incidence. While I understand your reason, I believe it would be beneficial to acknowledge this limitation in the discussion section.

Response: Thank you for your suggestion. The text has been revised, and a sentence addressing the age range limitation has been included in the discussion section under the limitations paragraph, acknowledging the gap between 5 and 15 years old and its relevance to studies on caries incidence at 6 and 12 years old.

Revised text:

Furthermore, it is important to note that the age range was determined based on the availability of comparable data. The gap between 5 and 15 years, particularly given the common focus on 6- and 12-year-olds in caries incidence studies, may affect comparing the findings of the present study with those of oral epidemiology surveys that used the ages recommended by the WHO pathfinder oral health surveys method.

RESULTS:

-In line 216 I think it's important to reinforce that you are still talking about 2019 data to make it easier for the reader to find the number in the table. The same comment is valued for lines 218, 220, 221 and 223.

Response: Thank you for your suggestion. The text has been revised, and we chose to rewrite certain sections to make it clearer that the information refers to the year 2019.

Revised text:

The results of the analysis of untreated caries in deciduous teeth revealed a social gradient, with more developed countries showing smaller prevalence, incidence and YLDs. Table 1 shows the prevalence of untreated caries in primary dentition by structural determinants of health in 2000, 2010 and 2019. In 2019, high GDP countries had a mean untreated deciduous dental caries prevalence of 32,373.3/100,000 (95% CI [29,561.5-35,185.2]), whereas middle and low GDP countries had mean prevalence of 40,587.2/100,000 (95% IC [39,514.5-41,660.0]) and 36,573.1/100,000 (95% CI [35,313.0-37833.1]), respectively. In 2019, differences were observed in the prevalence of health expenditures, with countries in the lower and middle tertiles presenting the highest mean prevalence of untreated caries, 38,384.6/100,000 (95% CI [36,814.8 - 39,954.4]) and 37,204.0 (95% CI [35,792.3- 38,615.6]), respectively. The upper tertile had a mean prevalence of 32,755.7 (95% CI [30,145.0-35,366.5]).

-In line 218 you say Regarding the Social Development Index (SDI), the third quintile presented the highest mean prevalence, 220 39,319.1/100,000 (95% CI [37,903.3 – 40,734.9]), while the upper quintile presented the smallest mean prevalence, 30,289.9/100,000. (95% CI [26550.2-34029.5]). The possible reason for this has to be in the discussion session.

Response: Thank you for your comment. A section addressing a possible explanation for the higher prevalence of caries in the third quintile of the SDI was added to the fourth paragraph of the discussion. It highlights the impact of economic and nutritional transitions, the increased consumption of ultra-processed foods, and inequalities in access to oral health services across countries.

Revised text:

Middle-income countries also suffer more from the influence of large corporations and commercial drivers, contributing to the global increase in the consumption of foods high in sugar.34,35 Additionally, food insecurity, which is more prevalent in these countries, often leads to a greater reliance on sugar-based foods due to their affordability, further exacerbating tooth decay and oral health disparities.35 These dynamics may partially explain why countries in the third quintile of the SDI, despite having more resources than those in the lowest quintile, still have higher prevalence of caries.36 This suggests that factors such as rapid nutrition transitions within the middle-income category. 36 This aspect emphasizes the need for an intersectoral approach based on a health promotion perspective to these countries.36This aspect emphasizes the need for a broader approach to oral health policies in low- and middle-income countries.36

References:

35. Sheiham A, James WPT. A new understanding of the relationship between sugars, dental caries and fluoride use: implications for limits on sugars consumption. Public Health Nutr. 2014;17(10):2176-2184. doi:10.1017/S136898001400113X

36. Kiosia A, Dagbasi A, Berkley JA, et al. The double burden of malnutrition in individuals: Identifying key challenges and re‐thinking research focus. Nutrition Bulletin. 2024;49(2):132-145. doi:10.1111/nbu.12670

-In line 219 you mention the Social Development Index (SDI) but in the table is written Sociodemographic index (SDI). Which one is correct?

Response: Thank you for the observation. The term for SDI is Sociodemographic index. It has been standardized throughout the text.

-I think it's important that line 227 continues by noting that, for most structural determinants, YLDs increase over time. This point should be discussed further later in the manuscript.

Response: Thank you for this suggestion. We have added this information in the results section and discussed it in the discussion session, as follows:

Results:

In respect to the variable ‘women with seats in national parliaments’, countries where women occupied more seats had lower mean deciduous caries prevalence, at 33,233.0/100,000 (95% CI [30,749.2-35,716.9])], when compared to countries with fewer women with seats in parliament, with a mean prevalence of 37,926.5/100,000 (95% CI [36,776.5-39,076.5]). A similar pattern is visible in 2000, 2010 and 2019 for the incidence and YLDs. For most structural determinants, YLDs increased over time. The social gradient was spanned over the two-decades period analyzed (Tables 1 and 2).

Discussion:

Another possible explanation for the higher YLDs observed in primary teeth, compared to permanent teeth, is the lower likelihood of restorative treatment in the primary dentition. In many settings, primary teeth are perceived as temporary and less important, which may lead to under-treatment. For instance, a study conducted among parents in Chennai, India, found that 68% believed treatment of primary teeth was unnecessary because they would eventually shed, and 43% thought that missing spaces in primary dentition could be left unaddressed until the eruption of permanent teeth.31 This perception contributes to reduced clinical attention and prioritization, resulting in a greater burden of untreated caries and, consequently, higher disability estimates in early childhood. These patterns may also reflect structural barriers to timely care, particularly among children at higher risk of caries.

Reference:

31. Ramakrishnan M, Banu S, Ningthoujam S, Samuel V. Evaluation of knowledge and attitude of parents about the importance of maintaining primary dentition - A cross-sectional study. J Family Med Prim Care. 2019;8(2):414. doi:10.4103/jfmpc.jfmpc_371_18

-Why did the authors not provide data in the third

---

## [Decision Letter · Decision Letter 1]

7 May 2025

Structural determinants of inequalities in untreated dental caries in the Global Burden of Disease Study

PONE-D-25-05748R1

Dear Dr. Hugo,

We’re pleased to inform you that your manuscript has been judged scientifically suitable for publication and will be formally accepted for publication once it meets all outstanding technical requirements.

Kind regards,

Thiago Machado Ardenghi

Academic Editor

PLOS ONE

Additional Editor Comments (optional):

You have made good work in revising Your manuscript and You seem to have carefully observed the comments and developed Your paper according to the reviewer suggestions.

Reviewers' comments:

Reviewer's Responses to Questions

**Comments to the Author**

1. If the authors have adequately addressed your comments raised in a previous round of review and you feel that this manuscript is now acceptable for publication, you may indicate that here to bypass the “Comments to the Author” section, enter your conflict of interest statement in the “Confidential to Editor” section, and submit your "Accept" recommendation.

Reviewer #2: All comments have been addressed

2. Is the manuscript technically sound, and do the data support the conclusions?

Reviewer #2: Yes

3. Has the statistical analysis been performed appropriately and rigorously? 

Reviewer #2: Yes

4. Have the authors made all data underlying the findings in their manuscript fully available?

Reviewer #2: Yes

5. Is the manuscript presented in an intelligible fashion and written in standard English?

Reviewer #2: Yes

6. Review Comments to the Author

Reviewer #2: Dear Authors,

Congratulations once again on this beautiful study. I appreciate your thoughtful responses to the issues I previously raised.

The only remaining concern is the non-functional link on line 120, which appears to still be inactive.

It is my pleasure to recommend acceptance of this study.

Best regards,

7. PLOS authors have the option to publish the peer review history of their article (what does this mean? ). If published, this will include your full peer review and any attached files.

**Do you want your identity to be public for this peer review?** For information about this choice, including consent withdrawal, please see our Privacy Policy .

Reviewer #2: **Yes: ** Ana Luiza Peres Baldiotti, DDS, MSc, PhD

---

## [Editor Report · Acceptance letter]

PONE-D-25-05748R1

PLOS ONE

Dear Dr. Hugo,

I'm pleased to inform you that your manuscript has been deemed suitable for publication in PLOS ONE. Congratulations! Your manuscript is now being handed over to our production team.

Kind regards,

on behalf of

Dr. Thiago Machado Ardenghi

Academic Editor

PLOS ONE